# Context-Aware Ubiquitous Learning Based on Case Methods and Team-Based Projects: Design and Validation

**I Kadek Suartama** [1,*] , **Eges Triwahyuni** [2] **and Kadek Suranata** [3]

1   Department of Educational Technology, Faculty of Educational Science, Universitas Pendidikan Ganesha, Singaraja 81116, Indonesia
2   Department of Educational Technology, Faculty of Educational Science, Institute Teacher Training and Education of PGRI Jember, Jember 68111, Indonesia
3   Department of Guidance and Counseling, Faculty of Educational Science, Universitas Pendidikan Ganesha, Singaraja 81116, Indonesia
*   Correspondence: ik-suartama@undiksha.ac.id; Tel.: +62-819-1635-8566

**Abstract:** Since the development of information and communication technology 'learning media' have been recognized as increasingly important courses by educators to assist students training to be professional teachers to understand the process of designing, developing, utilizing, managing, and evaluating learning resources. The context-aware ubiquitous learning approach design based on case methods and team-based projects has been proposed to guide activities in the student learning process. Therefore, this research aims to describe the design and determine the feasibility or validity of context-aware ubiquitous learning based on case methods and a team-based project model using the research and development (R&D) approach. The procedure consists of (1) model development and (2) product validation. It is important to note that the validation aspect involved five learning material/design experts, five 'learning media' experts, and 162 students from state and private universities in Indonesia after which the data obtained from experts and students were analyzed using the descriptive statistics. The results showed that the learning model designed is suitable for use in the student learning process.

**Keywords:** context-aware ubiquitous learning; case methods; team-based projects; learning media

## 1. Introduction

The competencies of students need to be more compatible with societal needs in order to adapt to the changes in the social and cultural space, the world of work, and the rapid technological advancement. This implies the focus should not be placed only on aligning their education with the industry and world of work but also to ensure they adapt to the rapidly changing future [1]. Therefore, universities are required to design and implement innovative and adaptive learning processes to achieve outcomes covering the aspects of attitudes, knowledge, and optimal skills as well as to ensure the students keep up with relevant innovations.

'Learning media' is one of the compulsory courses for students enrolled in educational study programs at tertiary institutions categorized as Educational Personnel Education Institutions charged with producing professional teacher candidates in Indonesia [2,3]. It was designed in an effort to respond and adapt to the recent rapid development and progress of information and communication technology (ICT) and to create different resources to be used in teacher training. These are also intended to support the formation of pedagogic competencies for potential teachers, especially in terms of designing, developing, utilizing, managing, and evaluating learning resources or media.

An analysis of documentation studies, surveys, and observations, as well as personal experiences of those involved in teaching 'learning media' courses has provided some data and information. First, the course has a fairly wide range of material consisting of

theoretical concepts as well as practical and empirical materials. The competencies to be achieved include the knowledge, skills, and attitudes in the fields of design, development, utilization, management, and evaluation or study of 'learning media'. It is weighted with three credits with each equivalent to 50 minutes and this signifies the time allocated to study in class is relatively small compared to the extensive material developed for the course. Therefore, a learning strategy is needed that allows students to be able to learn anytime, anywhere, and independently outside of the formally allocated time. Second, the final score for the 'learning media' course in 2021 showed that 1 student (2%) had C, 3 (3%) had C+, 5 (4%) had B-, 19 (10%) had B, 62 (35%) had B+, 60 (33%) had A-, and only 27 students represented by 14% had A grade. In the learning system approach, the low student learning outcomes indicate that learning components such as instrumental, process, and learning environments are not running optimally. These non-optimal results reflect the low quality of learning caused by the poor quality of activities. Third, almost all the students that participated in the course already had the ability to operate information and communication technology devices and possess laptops, tablets, and smartphones with some even having more than one device. However, based on previous research, Ref. [4] found that students mostly use these devices for entertainment, to play games, and use social media with only a few using them to seek information, study, collect assignments, and discuss. The fourth is related to environmental and infrastructure support and the campus was observed to already have several computer laboratory units with adequate numbers of PCs, Wi-Fi for internet connection, and almost all lecturers and students had information and communication technology tools and the capacity to use them. It was also observed that most of the lecturers and students subscribed to cellular data packages to have internet access outside the campus area.

Online learning activities increased significantly when the COVID-19 pandemic started [5] but this led to several problems due to the unpreparedness of both the lecturers and students. The challenges faced by the lecturers were reported by Irfan, Kusuman-ingrum, Yulia, and Widodo [6] to be caused by the presentation of monotonous and boring materials as well as the difficulty in being active and creating interaction with their students and between the students. It was also observed that online learning was more stressful than studying in a regular classroom because students were stuck studying alone and the absence of authentic activities made it difficult to focus on learning [7]. Moreover, online contents are theoretical and did not provide opportunities for students to practice, learn meaningfully, and study effectively [8]. This led to an incomplete learning process using these platforms [9].

The present industrial revolution 4.0 era has massively increased our dependence on information and communication technology to the extent that ICT has become the main basis of human life. In particular, the COVID-19 pandemic ushered in the new normal era or the adaptation of new habits and caused several changes in educational and learning practices [10,11]. Examples of these include making work and studying from home effective and successful while being interesting, challenging, and fun. It is important to note that the availability of technology such as the internet, various online meeting applications, tablets, smartphones, a computing software, graphics, animation, visuals, audio, games, and other digital devices provide new opportunities to develop meaningful online learning strategies which can be used anytime, anywhere, and in different ways.

These conditions served as the driving force to develop and implement the learning that can be completed anytime and anywhere using different methods in relation to the modalities or preferences of students. This approach considered one of the relevant solutions to several problems including those associated with 'learning media' courses. An innovative learning approach perceived to be effective is ubiquitous learning which is defined as any scenario where students are allowed to be fully immersed or involved in the learning process [12,13]. Such learning environments allow the creation of more active and adaptive learning activities in the real world. They provide the students with the opportunity to learn at the right time, in the right place, with the right content, and using

the right device [14]. Ubiquitous learning works on three main resources which include learning collaborators, contents, and services [15–17]. Some of its characteristics include context-awareness, interactivity, personalization, and flexibility [18].

Contextualized and context-aware learning refers to active strategies to utilize context in studying design [19]. Context-awareness is a strategy that adapts students to their environment by capturing and understanding their characteristics. It is an authentic learning strategy supported by personalized digital technology which allows students to observe or study real-world objects and activities through digital guidance [20]. This strategy ensures access to specific learning resources, content, or interactive activities based on the location, time, activity, and environment of the students. It also supports individual and group learning as well as content and information management while providing instruction and feedback based on the time, location, or activity conducted by the students. Context-awareness learning strategies allow seamless learning from one place to another within a predetermined area [14]. Moreover, context-awareness consist of ten context components which include personal, task, device, social, spatio-temporal, environmental, user interface, infrastructure, strategic, and historical contexts. It can be built using a web-based learning management system (LMS) using different features to include many functions [21].

It is also possible to apply several student-centered learning methods to implement the principles of context-aware ubiquitous learning [22] such as case [23] and team-based project learning methods [24]. The case method is constructivist learning which involves presenting real problems close to the students' lives. Ali et al. [25] stated that it allows the students to (1) analyze cases and content, (2) increase exploratory knowledge through the independent exploration of information, data, and literature, (3) improve critical thinking by solving the cases provided, (4) achieve better collaboration through discussions to find answers, and (5) increase the opportunity to receive feedback through presentations and improvements. The cases presented usually have problems related to the environment, conditions, situations, or a picture associated with the future of the student [26].

The team-based project involves using projects and activities as the learning medium such that the students are expected to explore, assess, interpret, and synthesize information to produce different forms of learning outcomes [27]. It is a student-centered learning model usually applied to investigate a topic in depth by ensuring the students apply collaborative and constructive deep learning with a research-based approach to serious, real, and relevant problems and questions [28]

The learning innovations developed based on context-aware ubiquitous learning strategies through case methods and team-based projects are considered relevant to technological developments in the industrial revolution 4.0 era. This is due to the fact that this model allows seamless learning anywhere and anytime according to the preferences and environment of the students. It also ensures an uninterrupted learning process while moving from one place to another and has the ability to connect, integrate, and share learning resources in the right place at the right time through an interoperable, pervasive, and seamless learning environment.

Therefore, the objectives of this research are to (1) describe the model and (2) determine the feasibility of the context-aware ubiquitous learning model, designed based on case methods and team-based projects for 'learning media' courses.

## 2. Materials and Methods

### 2.1. Research Design

Learning design models can be developed theoretically, practically, or with both methods. The theoretical approach works by synthesizing related literature while the practical method utilizes a simulation or a learning activity design project [29,30]. The model used in this study was developed by synthesizing relevant literature, simulating design tasks, and implementing a learning activity project.

Learning design models can be classified into three stages which include (1) development, (2) validation, and (3) use [31]; however, this research focuses on two stages—development and validation. It is important to note that the instructional design model validation is a carefully planned process of collecting and analyzing empirical data to (1) provide support for each component of the model or (2) prove its usefulness in practice [32]. It can be conducted internally, externally, or through both methods after which the model can be tested or implemented in learning activities.

### 2.2. Procedure

#### 2.2.1. Model Development

The first step in the research design adopted is model development. The context-aware ubiquitous learning based on case methods and the team-based project design model was developed in line with the model proposed by Hsu and Hwang [33]. The structure consists of a registration and authorization module, a learning sequence or step guide module, a management system, a student account database, a learning material database, and a learning portfolio database. The learning system design model is guided by basic pedagogical theories, especially case methods [25] and project-based learning [34], and supported by learning management system technology [35] as shown in the following Figure 1.

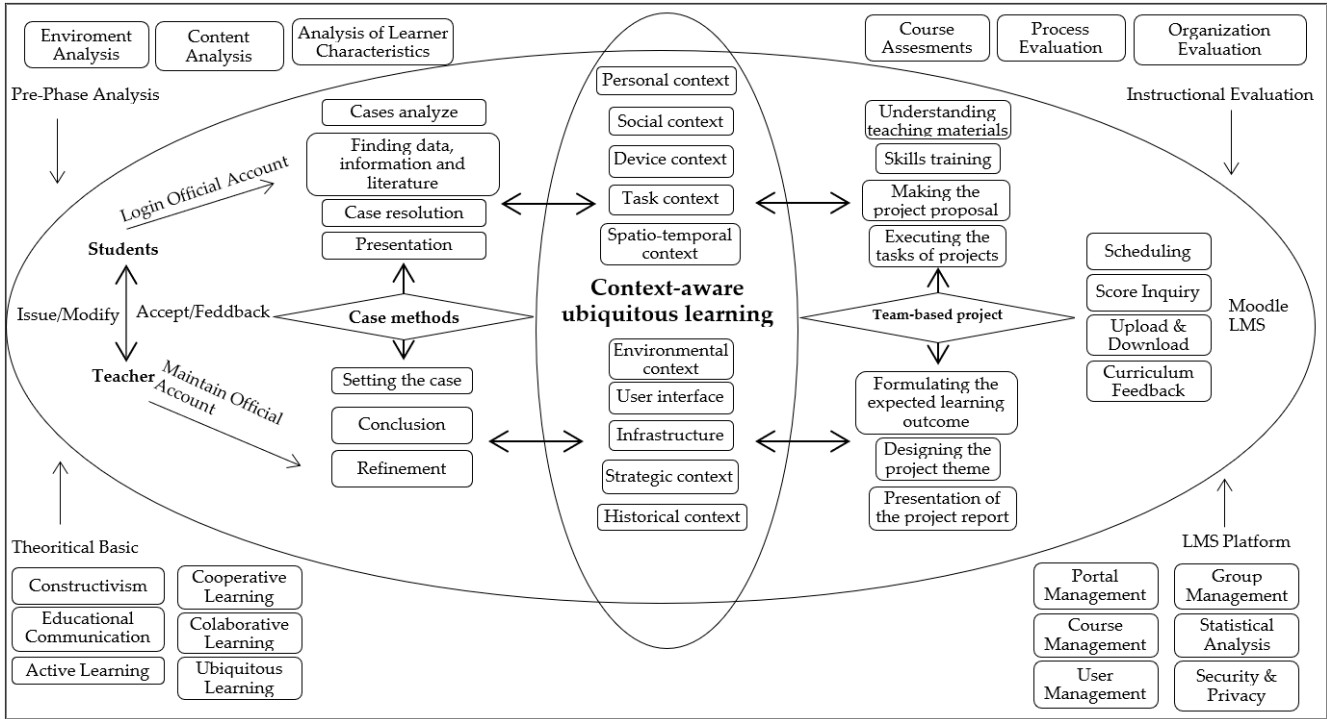

**Figure 1.** The structure of the context-aware ubiquitous learning design model based on case methods and team-based projects.

Figure 1 shows the design structure of the model which combines the pedagogical and technological aspects. A context-awareness learning environment can be built using a web-based learning management system (LMS) using different approaches to include many functions. This model allows students to access specialized learning resources, content, and interactive activities at their convenience. It also supports information, content management, and individual activities as well as the provision of instructions and feedback in line with the situation, environment, and desires of the students. The development of an authentic learning environment supported by personalized digital technology is an effort to create context-awareness ubiquitous learning. This allows students to observe or classify real-world objects in learning activities through digital guidance. The model also ensures

the provision of seamless learning from place to place within a defined area. It is important to note that the ten principles of context awareness were implemented in each step of the case method and team-based project learning.

Case-based learning (CBL) is closely related to realistic problem scenarios relevant to the material being studied and involves the active participation of students to integrate several sources of information into context to solve cases based on previous experience and knowledge. The application of this method is characterized by several components or steps which include (1) determination of the case, (2) analysis of the case, (3) independent exploration of information, data, and literature, (4) determination of the steps to solve the case, (5) conclusion of the answers discussed together, (6) presentation, and (7) improvement.

The team-based project method is, however, characterized by several components or steps which involve (1) formulation of the expected outcome, (2) understanding the concept of the teaching materials, (3) skills training, (4) designing the project theme, (5) making the project proposal, (6) execution of the project task, and (7) presentation of the project report.

The next step based on the design is the development which involves translating the design into a physical form such as a portal and an e-learning course. The portal creation process includes (1) obtaining a server/web hosting, (2) changing the identity of the portal such as the site name and description, (3) setting up the Moodle mobile app, (4) changing the theme, (5) creating categories, and (6) creating and uploading user status. Meanwhile, the course creation stages include (1) creating and changing course settings, (2) uploading resources such as books, files, folders, IMS content packages, labels, pages, and URLs, and (3) creating activities such as assignments, chat, choice, database, feedback, forum, glossary, lesson, LTI/external tool, quiz, SCORM, survey, wiki, and workshop.

2.2.2. Product Validation

Validator

The product developed was validated by 5 lecturers teaching media courses and 5 experts in 'learning media' with the appropriate educational background and significant experience in their fields such as doctors in educational technology. This expert validation is important to improve and guarantee the feasibility of implementing the context-aware ubiquitous learning model based on case methods and team-based projects in learning.

Trial Subject

In addition to the experts, the product was also tested on 162 students to determine its usefulness and quality before being used in actual learning. The students were asked to fill out the product assessment questionnaire after which their responses were analyzed, and the findings were then used for revision before the final product was produced. This trial process is expected to ensure better quality for the products developed.

Instrument

The instrument used to validate the learning material or design is a questionnaire developed by Walker and Hess [36] while the online 'learning media' aspect was assessed through the comprehensive standard or rubric questionnaire for online learning design developed by Debattista [37]. The questionnaires used during the trial activities were prepared and evaluated by the material or design and media experts by adjusting the statements or questions to suit the students as the users of the product. The items used are indicated in Tables 1 and 2.

**Table 1.** Grid of material assessment instruments and learning design.

| Aspect | Indicator |
|---|---|
| Material | The suitability of the material with the competencies to be achieved<br>Concept truth<br>Material updates<br>Order of material presentation<br>The suitability of the given example |
| Learning | Learning objectives<br>Motivation<br>Summary<br>Clarity of learning indicators<br>Giving training<br>Suitability of images, and videos provided to clarify the material |
| Language | The suitability of language with students' thinking level<br>Simple language<br>Accuracy of terms<br>Grammar and spelling accuracy<br>Ability to arouse students' curiosity |

**Table 2.** Grid of online 'learning media' assessment instruments.

| Main Standards | Specific Standards |
|---|---|
| Course opening | Behavior<br>Role<br>Accessibility<br>Integrity<br>Technical competences<br>Ownership |
| Instructional resources for teaching and learning | Openness<br>Provision<br>Entitlement<br>Application<br>Variety<br>Academic integrity |
| Interaction and community | Peer learning<br>Fostering<br>Management |
| Learner support | Academic<br>Instructional<br>Administrative<br>Technical |
| Technology design | Interface<br>Access<br>Centricity<br>Investment<br>Authentication<br>Management |
| Course closing | Conclusions<br>Archiving<br>Resolution |

**Table 2.** *Cont.*

| Main Standards | Specific Standards |
|---|---|
| Assessment of learning | Measurement<br>Grading<br>Management<br>Feedback |
| Instructional design cycle | Academic<br>Administrative<br>Technical |

Data Analysis Technique

The data obtained after the learning model had been validated were classified into 2 parts—qualitative and quantitative. The qualitative data were in the form of criticism and suggestions put forward by the experts to improve the learning model while the quantitative data were the scores provided for each item of the instrument. Moreover, descriptive statistical analysis techniques were applied to determine the value or quality of the digital learning system developed. The scores obtained were added up, averaged, and converted into scores using a 5-scale criterion-referenced test table adapted from Sukardjo [38] as presented in Table 3.

**Table 3.** Scores converted on a five scale.

| Value/Category | Score | |
|---|---|---|
| | Formula | Calculation |
| Very good | $X > \overline{X}_i + 1.80$ Sbi | $X > 4.21$ |
| Good | $\overline{X}_i + 0.60$ Sdi $< X \leq \overline{X}_i + 1.80$ Sdi | $3.40 < X \leq 4.21$ |
| Quite good | $\overline{X}_i - 0.60$ Sdi $< X \leq \overline{X}_i + 0.60$ Sdi | $2.60 < X \leq 3.40$ |
| Not good | $\overline{X}_i - 1.80$ Sdi $< X \leq \overline{X}_i - 0.60$ Sdi | $1.79 < X \leq 2.60$ |
| Bad | $X \leq \overline{X}_i - 1.80$ Sdi | $X \leq 1.79$ |

Description: Ideal average ($\overline{X}_i$) = 1/2 × (max score + minimum score); standard deviation ideal (Sdi) = 1/6 × (max score–minimum score); maximum score = 5; minimum score = 1; $\overline{X}_i$ = 1/2 × (5 + 1) = 3; Sdi = 1/6 × (5 − 1) = 0.67; X = actual score.

A minimum feasibility value of "Good" was obtained in this research based on the assessments of the experts. It is important to note that the achievement of at least a "good" score for the final (overall) assessment by the experts indicates the learning design model developed is considered feasible to be applied in learning.

## 3. Results

*3.1. Context-Aware Ubiquitous Learning Model Design Based on Case Methods and Team-Based Projects for the 'Learning Media' Course*

The components of the learning model design including the materials/topics, methods, types of teaching materials, resources, and activities were organized in the form of scenarios or a mapping program. The mapping program is a table containing the materials/topics, sequence of learning steps/methods, types of teaching materials, resources, and activities for one semester with a link connected to the complete material integrated into each element [35]. The benefit of this mapping program includes ease and accuracy in connecting learning elements to develop the course. Moreover, the creation of a course requires preparing the learning materials in digital formats such as documents (doc, pdf, xls, txt), presentations (ppt), images (jpg, gif, png), videos (mp4, mpg, wmv), sound (mp3, au,

wav), and animation (swf, gif). It is important to reiterate that the methods used were case methods and team-based projects. Therefore, the Moodle LMS feature consists of the resources (books, files, folders, ims content packages, labels, pages, and URLs) and activities (assignments, chat, choice, database, feedback, forums, glossary, lessons, lti/external tools, quizzes, scorms, surveys, wikis, workshops). The last element is related to the implementation of context-aware ubiquitous learning principles which include the personal, task, device, social, spatio-temporal, environmental, user interface, infrastructure, strategic, and historical contexts. These are, therefore, presented in the following Table 4.

**Table 4.** Mapping context-aware ubiquitous learning program based on case methods and team-based projects for the 'learning media' course.

| Topics | Method | Type of Teaching Material | Moodle LMS Features | | Implementation of Context-Aware Ubiquitous Learning |
|---|---|---|---|---|---|
| | | | Resource | Activities | |
| Position of 'learning media' | Case methods | Document (pdf) | Page File URL | Forum Assignments Feedback | Personal context Social context Spatio-temporal context Environmental context Historical context |
| The basic concept of 'learning media' | Case methods | Presentation (ppt) | Page File URL | Forum Assignments Feedback | Personal context Social context Spatio-temporal context Environmental context Historical context |
| Classification of 'learning media' | Team-based projects | Videos (mp4) | Book File URL | Assignments Chatting Feedback | Task context Device context User interface Infrastructure Strategic context |
| Characteristics of various types of 'learning media' | Case methods | Document (pdf) | File URL | Lesson BigBlueButtonBN (web conference) Messages | Personal context Social context Spatio-temporal context Environmental context Historical context |
| Learning media management | Case methods | Presentation (ppt) | Label Page | Workshop Assignments | Personal context Social context Spatio-temporal context Environmental context Historical context |
| Selection of 'learning media' | Case methods | Videos (mp4) | Page File URL | Forum Assignments Feedback | Personal context Social context Spatio-temporal context Environmental context Historical context |
| Analysis of 'learning media' needs | Team-based projects | Documents (doc, pdf) | Page File URL | Forum Assignments Feedback | Task context Device context User interface Infrastructure Strategic context |
| 'Learning media' design and production | Team-based projects | Presentation (ppt) | Page File URL | Forum Assignments Feedback | Task context Device context User interface Infrastructure Strategic context |

**Table 4.** *Cont.*

| Topics | Method | Type of Teaching Material | Moodle LMS Features | | Implementation of Context-Aware Ubiquitous Learning |
|---|---|---|---|---|---|
| | | | Resource | Activities | |
| Evaluation of 'learning media' | Team-based projects | Images (jpg, png) | Page File URL | Forum Assignments Feedback | Task context Device context User interface Infrastructure Strategic context |
| Use of 'learning media' | Case methods | Videos (mp4) | Page File URL | Forum Assignments Feedback | Personal context Social context Spatio-temporal context Environmental context Historical context |
| Simple media production practice | Team-based projects | Document (pdf) | Page File URL | Forum Assignments Feedback | Task context Device context User interface Infrastructure Strategic context |
| Digital 'learning media' production work practice | Team-based projects | Presentation (ppt) | Page File URL | Forum Assignments Feedback | Task context Device context User interface Infrastructure Strategic context |

### 3.2. The Final Product

The final product is an e-learning course developed based on context-aware ubiquitous learning through the case and team-based project methods for 'learning media'. It consists of 15 segments/sections which are stated as follows:

(1)    Introduction to courses
(2)    Learning activities 1: The position of 'learning media'
(3)    Learning activities 2: Basic concepts of 'learning media'
(4)    Learning activities 3: Classification of 'learning media'
(5)    Learning activities 4: The characteristics of the types of 'learning media'
(6)    Learning activities 5: 'Learning media' management
(7)    Learning activities 6: Selection of 'learning media'
(8)    Mid-semester exam
(9)    Learning activities 7: Analysis of 'learning media' needs
(10)    Learning activity 8: Design and production of 'learning media'
(11)    Learning activity 9: Evaluation of 'learning media'
(12)    Learning activities 10: Use of 'learning media'
(13)    Work practice 1: Production of simple 'learning media'
(14)    Work practice 2: Digital 'learning media' production
(15)    Final exams

There are six activity segments designed through the case study method and these include the learning activities 1, 2, 4, 5, 6, and 10. The topics are very relevant to the students because they cover realistic problems associated with their environment, conditions, situations, or a description of their future. The application of this method is facilitated by interactive guidance/student worksheets (LKM-case method). Moreover, the steps involved in line with the context-aware ubiquitous learning principles are stated as follows:

(1)    Determine cases based on the environmental context by selecting issues associated with using 'learning media' in schools close to the student environment
(2)    Analyze cases with the principle of social context by identifying or perceiving cases based on the closest social environment

(3)    Independently find information, data, and literature based on personal context by using the device brought individually in line with their interests, motivations, knowledge, and concerns

(4)    Determine the steps to solve the cases through spatio-temporal context by applying spatial-temporal reasoning to solve multi-step problems

(5)    Discuss and conclude the answers together using the historical context by paying attention to the social, political, cultural, economic, and environmental situations affecting the events or trends observed at a certain time.

(6)    Present the findings through the spatio-temporal, environmental, and historical contexts

(7)    Improve on the findings based on the personal and social contexts

There are six activity segments designed to be implemented through the team-based project method and these include learning activities 3, 7, 8, and 9, as well as work practices 1 and 2. The topics are very relevant to the students in ensuring they have the ability to provide a hands-on experience (skills in developing 'learning media') collaboratively and present learning that is not limited to mere knowledge (cognitive). This method is also facilitated by interactive guides/student worksheets (LKM-team-based projects). The steps in relation to the context-aware ubiquitous learning principles are as follows:

(1)    Formulate the expected learning outcome based on the task context by having a set of special conditions characterizing the situation related to the task. This assists in adjusting the resource capabilities and directing efforts to better fit the situation in order to reduce inefficiencies and avoid several potential errors.

(2)    Understand the concept of the teaching materials based on personal context through the provision of subject matter by considering the preferences, interests, motivations, knowledge, and concerns of each individual.

(3)    Develop skills based on the device and infrastructure contexts by providing different alternative exercises based on the device owned by each individual

(4)    Design the project theme based on the task context

(5)    Produce the project proposal through the task, personal, device, and infrastructure contexts

(6)    Execute the tasks of the projects based on the strategic context by ensuring accuracy in the steps to achieve the stated goals

(7)    Present the project report based on the spatio-temporal, environmental, and historical contexts

The complete 'learning media' course can be accessed through the guest login feature at the following address: https://u-learningclass.site/course/view.php?id=2 (accessed on 15 August 2022).

### 3.3. Product Validity

The next process was internal testing of the product to ensure it runs smoothly and this was followed by the evaluation stage which includes (1) validation by five material/design experts that are teaching media courses from five different universities and five media experts that are doctors in the field of learning technology as well as (2) the application of the product to 162 students from the state (Universitas Pendidikan Ganesha, Open University) and private universities (IKIP PGRI Jember) in Indonesia.

The assessment by the material/design experts produced the results presented in the following Table 5.

**Table 5.** Material expert validation results for the product developed.

| Assessment Aspect | Average Score | Category |
| --- | --- | --- |
| Material Aspect | 4.32 | Very good |
| Learning Aspect | 4.23 | Very good |
| Language Aspect | 4.24 | Very good |
| Average Overall Score | 4.26 | Very good |

Table 5 shows that the average score for the overall assessment of the learning material/content aspects was 4.26 which, according to the conversion table of quantitative to qualitative data on a scale of 5, belongs to the "Very Good" category. This implies the learning materials developed are feasible to be applied in learning activities.

The suggestions for revision and improvement made by the material/learning design experts are as follows (1) it is necessary to add learning/introduction instructions to each learning activity to assist students in self-study, (2), enrich the 'learning media 'material based on digital/ICT technology to ensure it is updated, (3) the presentation of the material should allow students to think critically by presenting factual problems related to the non-optimal application of 'learning media' in schools, and (4) add more tutorials on using different authoring tool applications to create digital 'learning media' in order to develop the materials.

The assessment made by media experts on the product quality is also presented in the following Table 6.

**Table 6.** Media expert validation results for the product developed.

| Assessment Aspect | Average Score | Category |
|---|---|---|
| Opening | 4.83 | Very good |
| Learning resources | 4.67 | Very good |
| Interaction and communication | 4.67 | Very good |
| Student/study support | 4.70 | Very good |
| Technology design | 4.73 | Very good |
| Closing/course closing | 4.73 | Very good |
| Evaluation | 4.80 | Very good |
| Learning cycle | 4.60 | Very good |
| Average Overall Score | 4.72 | Very good |

Table 6 shows that the average score for the overall assessment of the 'learning media' aspects is 4.72 which, according to the conversion table of quantitative to qualitative data on a scale of 5, belongs to the "Very Good" category. This also indicates the 'learning media' developed is feasible to be applied in learning activities.

The comments and suggestions for revision/improvement by the 'learning media' experts include: (1) the variety of learning resources needs to be increased to accommodate different student learning modalities, (2) a relevant and personal learning path is demanded to increase student interest in learning, (3) game elements such as progress bars, badges, points, and others should be considered to elevate student motivation in learning, and (4) feedback features are required to improve performance which can further assist the students to rethink.

The assessment results obtained from the trial activities conducted on 162 students are presented in the following Table 7.

Table 7 shows that the average score for the overall assessment of the online learning module aspects in the 'learning media' course is 4.37 which, according to the conversion table of quantitative to qualitative data on a scale of 5, belongs to the "Very Good" category. This further indicates the product developed is feasible to be applied in learning activities.

The comments from students generally indicate that (1) learning becomes more flexible because the course can be accessed anytime and anywhere, (2) assignments are clearly described and this is very helpful in ensuring they are completed, (3) assignments are given in groups, thereby, allowing the students to exchange experiences and complement each other, (4) the existence of group projects allows students to apply the theory and knowledge

gained, and (5) the course is easier to use and highly accessible, thereby, making learning more efficient.

**Table 7.** Students' assessment of the product through trials.

| Assessment Aspect | Average Score | Category |
|---|---|---|
| Material Aspect | 4.35 | Very good |
| Learning Aspect | 4.44 | Very good |
| Language Aspect | 4.38 | Very good |
| Opening | 4.40 | Very good |
| Learning resources | 4.39 | Very good |
| Interaction and communication | 4.38 | Very good |
| Student support | 4.32 | Very good |
| Technology design | 4.29 | Very good |
| Closing/course closing | 4.36 | Very good |
| Evaluation | 4.38 | Very good |
| Learning cycle | 4.34 | Very good |
| Average Overall Score | 4.37 | Very good |

The comparison of the average overall scores recorded from experts' validations and student trials is presented in the following Figure 2.

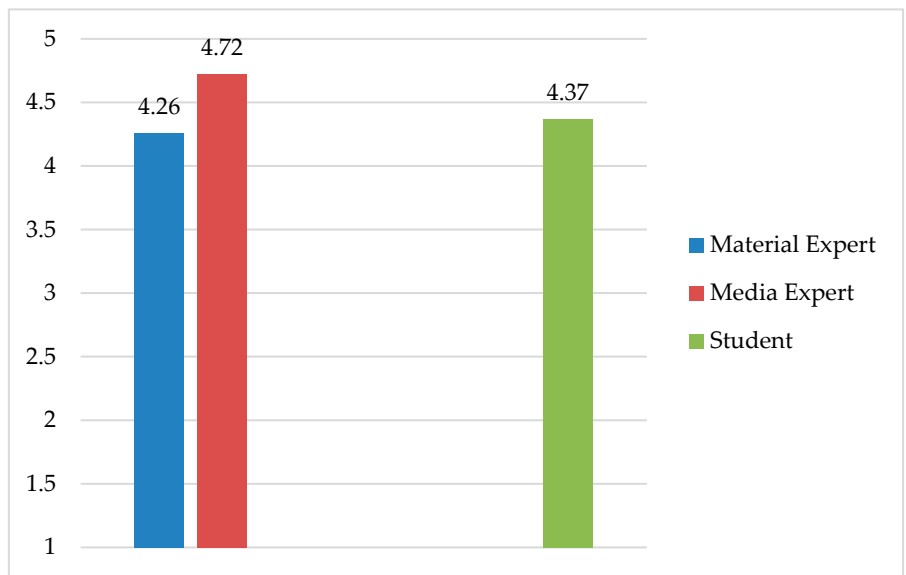

**Figure 2.** Comparison of results from product validation and trial.

Figure 2 shows that the average overall assessment score for the learning material/content aspect by five material experts is 4.26, the media aspect by five media experts is 4.72, and the average assessment score of 162 students in the product trial is 4.37. These figures, according to the table used in converting quantitative to qualitative data on a scale of 5, are in the "Very Good" category and this means the material and media aspects of the product developed are feasible to be applied in learning activities.

The suggestions, input, and comments from experts and students led to some improvements which include (1) the provision of a brief introduction to deliver the learning scenarios which include the learning objectives, resources, and the activities to be completed in each activity, (2) the addition of learning resources, both by design and utilization,

(3) the addition of problems regarding 'learning media' sourced from research results, especially in the presentation of case study methods, (4) the presence of video tutorials on the use of digital 'learning media' development applications, (5) the inclusion of lesson features in several segments to accommodate students with different learning speeds, (6) the addition of gamification features by installing gamification plugins on LMS such as Progress Bars, Badges, and Experience Points (XP).

## 4. Discussion

The process of developing a context-aware ubiquitous learning model design using the case method and team-based projects for the 'learning media' course was completed through the stages of model development and product validation. The model can run smoothly, quickly, and more orderly because it was established on a pre-made plan while the readiness of the required materials was obtained from the analysis conducted.

The validity assessment by material and media experts as well as the trials conducted on the students showed that the product is in a very good category. This indicates the design of the context-aware ubiquitous learning model using the case method and team-based projects for the 'learning media' course met the eligibility criteria for application in wider learning. It is feasible because the model combines varied and student-centered approaches.

The context awareness approach was implemented by providing field assignments to the students through an interactive guidance mechanism which was conducted digitally based on their location, time, and activities. This is necessary to assist the students in recognizing and distinguishing learning objects in the real world. The interactive guides also aid the collection and organization of the things observed and learned in the real world through the personal context. Moreover, the students' independent exploration of different learning resources, available in an unlimited environment and with social interactions conducted with different parties having direct or indirect relevance to the knowledge learned, indicate the application of the strategic, task, social, and environmental contexts. The learning goal was also achieved through the LMS features which are applied as assignments, chats, and feedback to represent the device context. The previous study by Wu, P.-H., Hwang, G.-J., and Tsai [39] showed that context-aware ubiquitous learning can significantly improve learning achievement on several cognitive processes in Bloom's taxonomy such as "analyzing" and "evaluating". It also indicated that the inclusion of an interactive guided approach in the model benefited students by improving their higher-order thinking competencies.

Activities in other learning segments are designed to support the historical context principle as indicated by the history or past experience of students. The use of the lesson feature also allows lecturers to deliver content and/or practicum activities in an interesting, flexible, and adaptive way, thereby, adjusting to the student achievement or learning outcomes. The lecturers create a series of pages of linear content or learning activities to offer the students different paths or options to ensure the students understand the concepts being taught through the provision of different true-false statements/questions. The choice of answers determines the next activity such that those with correct answers are allowed to proceed to the next activity while those with incorrect answers are returned to the previous page/enrichment material or path/discussion page. This enables the students to learn at their own pace in order to ensure optimal achievement of the learning outcomes [40].

Another method is for the lecturers to grade assignments and provide direct feedback to the students through the LMS. This feedback can either be a positive or negative reinforcement and this is related to the personal and device contexts. This is in line with the findings of previous studies that the existence of different assessment methods and access to grades usually spur students to continue studying hard to excel in class [41,42].

The context awareness approach facilitates students to access specific learning resources, content, and interactive activities based on their preferences and time. The concept supports information, content management, and individual activities while instructions

and feedback are provided based on the location, time, and activities. It also provides an authentic learning environment through case method and team-based projects through the support of digital technology and personalized activities. This allows students to observe real-world events and objects during the learning process through digital guidance [20]. Moreover, a context-awareness learning environment allows seamless learning from place to place within a defined area [14].

The other pedagogical elements such as the depth of the material, novelty, flow of the material presented, and linguistic aspect have also been fulfilled in the model design. This is due to the fact that it covers the course opening, instructional resources for teaching and learning, interaction and community, learner support, technology design, course closing, assessment of learning, and instructional design cycle. The other factors such as the provision of richer content or teaching materials, strategies, and a learning environment close to the student also have the ability to improve learning performance [43].

The application of the case-based method is a complex learning strategy closely related to realistic problem scenarios which are relevant to the material being studied. It allows the students to actively participate in integrating several sources of information into the context in an effort to solve cases based on their previous experience and knowledge [44]. This method is characterized by several components or steps which include (1) setting the case, (2) analyzing the case, (3) finding information, data, and literature independently, (4) determining the completion steps for the case, (5) discussing and concluding answers together, (6) presenting the findings, and (7) improving. This shows that the students do not only act as recipients of lessons explained verbally by the lecturers but also play a role in finding the core of the subject matter. The case-based learning method emphasizes student activities maximally as learning subjects by searching, finding, connecting, and applying concepts representatively in a low-risk environment [45].

There are several benefits for the students that complete this case-based learning process. It offers the opportunity to consider problems similar to those to be faced in the real world and to make connections between existing problems and the knowledge of the students [46]. The method also develops the "teamwork, critical thinking, and cultural awareness" skills of the students [47]. Moreover, all the activities conducted are directed toward finding answers to a particular concept and this allows the students to foster self-confidence and develop intellectual abilities as part of the mental process [48]. This simply means that case-based learning is effective for the development of problem-solving skills [49,50].

The team-based project method was applied to several other learning segments and has the ability to place the students in a role with greater autonomy, involvement, and responsibility for their learning activities [51]. The steps involved in this method include (1) formulating the expected learning outcome, (2) understanding the concept of the teaching materials, (3) skills training, (4) designing the project theme, (5) making the project proposal, (6) executing the tasks of projects, and (7) presenting the project report. It is important to note that all the project learning stages contribute to the improvement of critical thinking skills starting from the selection of the project that suits the wishes and needs of students. This is followed by the planning stage which begins with the students' prior knowledge as well as the setting and formulating of questions which enables the students to broaden their perceptions and thoughts on the activities built into project-based learning strategies. It also allows them to practice the skills of recognizing assumptions and evaluating arguments through class discussions. The next stages are project implementation, data collection and analysis followed by the evaluation phase and presentation of the project reports. These stages allow the students to connect their knowledge with real-life experience and also stimulate serious thinking when acquiring new knowledge [52].

The use of LMS features in the form of messages, chat, and forums is intended to create interaction between the lecturers and students as well as between the students during the project completion stage. This computer-mediated online social interaction is important in education due to some of its benefits such as flexibility and efficiency in terms of cost and

time. The findings of Araújo et al. [53] showed that social interaction and collaborative work in an ubiquitous learning environment can improve student performance. This implies interactive features enhance collaborative learning interactions as well as the teaching and learning process.

The design of the context-aware ubiquitous learning model using the case method and team-based projects for the 'learning media' course was based on the environment, preferences, and learning modalities of students. This is important because an effective learning environment is expected to consider the individual differences of the students because they learn in different ways and speeds according to their needs, interests, desires, and environment [54]. The model designed in this research provides an implementable, in-depth, and integrated learning architecture with three important components which include (1) collaborative learning, (2) learning activities or contents, and (3) learning services. The context-aware ubiquitous learning is characterized by the provision of an intuitive method of identifying the right collaborative learning, appropriate learning content, and adequate learning services at the right place and time [55].

## 5. Conclusions

The analysis and discussion showed that (1) the procedure to develop a context-aware ubiquitous learning model design using case methods and team-based projects in 'learning media' courses involves two stages of model development, and validation and (2) the design is feasible to be applied in learning.

## 6. Recommendations

The findings and discussion were used to formulate the following recommendations:

- This research product needs to be implemented among groups of university students taking 'learning media' courses.
- The lecturers should provide directions for students using this learning model for the first time even though it is designed for independent learning.
- The model design can be disseminated through different activities such as academic seminars organized by universities, training activities for the development of 'learning media', collaboration with learning centers, educational and training institutions, and other forums. It is designed and developed for several people interested in studying 'learning media'.
- The efforts to develop context-aware ubiquitous learning need to be implemented by the optimization of more varied presentation methods.
- It is necessary to pursue further research activities to identify the effectiveness of using this product through both classroom action research methods and experimental research in a wider target group.

## 7. Limitations

The development of this context-aware ubiquitous learning model design based on case methods and team-based projects for 'learning media' courses is limited in the following aspects:

- It needs to be applied using a computer or mobile device.
- It requires adequate internet access.
- It is limited to 'learning media' materials.
- No effectiveness test was conducted.
- The evaluation conducted did not consider the long-term impact.

**Author Contributions:** Conceptualization, I.K.S.; Data curation, K.S.; Formal analysis, K.S.; Investigation, K.S.; Methodology, I.K.S.; Project administration, I.K.S.; Resources, E.T.; Software, E.T.; Supervision, K.S.; Validation, E.T.; Visualization, E.T.; Writing—original draft, I.K.S.; Writing—review and editing, I.K.S. All authors have read and agreed to the published version of the manuscript.

**Funding:** This research was funded by the Directorate of Research, Technology and Community Service, Directorate General of Higher Education, Research, and Technology, Ministry of Education, Culture, Research and Technology of the Republic of Indonesia in accordance with the Contract for the Implementation of the National Competitive Research Program Number: 135/E5.PG.02.00.PT/2022.

**Institutional Review Board Statement:** Not applicable.

**Informed Consent Statement:** Not applicable.

**Conflicts of Interest:** The authors declare no conflict of interest.

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
