# Peer review of "Context-Aware Ubiquitous Learning Based on Case Methods and Team-Based Projects: Design and Validation"

_education, doi:10.3390/educsci12110802_

Round 1

Reviewer 1 Report

A very interesting and well-informed paper, with interesting and thoughtful recommendations. I found it a little challenging from the start and think the context could be clearer with some elaboration and illustration of what tertiary education and the media course and its students comprise. It is quite dense and I have made raised a few questions to follow.

Educational Personnel Education Institutions (LPTK) - is this initialism correct? Should education appear twice in the title?

“industrial revolution 4.0 {era]” is not properly explained, characterised or defined (nor argued that it is a real and present thing). 

Could learning media be rewritten ‘learning media’ to show it is a noun?

Not clear what is meant here: “Therefore, more time is 42 needed to complete learning anytime and anywhere”. Does this mean students need to dedicate themselves more to independent learning or that more time be formally allocated?

Please re-read and rewrite the paragraph that starts: "Contextualized and context-aware learning refer to an active approach to utilizing context in studying design. Context-awareness is a system that adapts students to their environment by capturing and understanding their characteristics. It is an authentic learning environment supported by personalized digital technology which allows students to observe or study real-world objects and activities through digital guidance. It is a concept..."

I found this unclear and it is a key concept to the paper, so must be expressed more plainly. Context-aware learning is within a few sentences various explained to be a concept, a platform, a system, an educational approach and a learning environment.  I'm not sure it can be all of the above and was not much better informed by the end where even more components were introduced to explain whatever it is. Perhaps an existing example of what it is and how it works would serve better to explain it.  

Reviewer 2 Report

This is interesting research but more clarity is needed in the introduction to guide the reader better on what follows. Include information relating to previous studies to position the research and its significance. The writing style and accuracy change throughout the paper, going from poor to very good. Specifically,

1. The abstract needs to be rewritten for better clarity.

2.  The methodology, results and conclusion are clear, but the structure of the Introduction needs to be improved. Begin with background information about the topic – a review of the literature and discuss the importance of the research by including a hypothesis, research questions, or research problem. Then outline the methodological approach and mention the key research findings. You should carefully link any previous studies discussed to the present work.

Your introduction should establish clearly the depth, context, and importance of your research by summarizing your thesis and bringing it to the attention of the reader, at the start of the paper.

3. See further comments in the attached document.

Round 2

Reviewer 2 Report

The writing has been improved but several grammatical issues are in need of attention. These are highlighted in the attached.
